# Derivation and Validation of Essential Predictors and Risk Index for Early Detection of Diabetic Retinopathy Using Electronic Health Records

**DOI:** 10.3390/jcm10071473

**Published:** 2021-04-02

**Authors:** Ru Wang, Zhuqi Miao, Tieming Liu, Mei Liu, Kristine Grdinovac, Xing Song, Ye Liang, Dursun Delen, William Paiva

**Affiliations:** 1Department of Statistics, Oklahoma State University, Stillwater, OK 74078, USA; ru.wang@okstate.edu (R.W.); ye.liang@okstate.edu (Y.L.); 2Center for Health Systems Innovation, Oklahoma State University, Tulsa, OK 74119, USA; dursun.delen@okstate.edu (D.D.); wpaiva@okstate.edu (W.P.); 3School of Industrial Engineering and Management, Oklahoma State University, Stillwater, OK 74078, USA; tieming.liu@okstate.edu; 4Division of Medical Informatics, Department of Internal Medicine, University of Kansas Medical Center, Kansas City, KS 66160, USA; meiliu@kumc.edu; 5Division of Endocrinology, Metabolism, and Genetics, Department of Internal Medicine, University of Kansas Medical Center, Kansas City, KS 66160, USA; kgrdinovac@kumc.edu; 6Department of Health Management and Informatics, University of Missouri, Columbia, MO 65212, USA; xsm7f@health.missouri.edu; 7Department of Management Science & Information Systems, Oklahoma State University, Tulsa, OK 74106, USA

**Keywords:** diabetic retinopathy, early detection, electronic health records, predictor selection, predictive models, risk index

## Abstract

Diabetic retinopathy (DR) is a leading cause for blindness among working-aged adults. The growing prevalence of diabetes urges for cost-effective tools to improve the compliance of eye examinations for early detection of DR. The objective of this research is to identify essential predictors and develop predictive technologies for DR using electronic health records. We conducted a retrospective analysis on a derivation cohort with 3749 DR and 94,127 non-DR diabetic patients. In the analysis, an ensemble predictor selection method was employed to find essential predictors among 26 variables in demographics, duration of diabetes, complications and laboratory results. A predictive model and a risk index were built based on the selected, essential predictors, and then validated using another independent validation cohort with 869 DR and 6448 non-DR diabetic patients. Out of the 26 variables, 10 were identified to be essential for predicting DR. The predictive model achieved a 0.85 AUC on the derivation cohort and a 0.77 AUC on the validation cohort. For the risk index, the AUCs were 0.81 and 0.73 on the derivation and validation cohorts, respectively. The predictive technologies can provide an early warning sign that motivates patients to comply with eye examinations for early screening and potential treatments.

## 1. Introduction

Diabetic retinopathy (DR) is a vision-threatening microvascular complication of diabetes, and is a leading cause of blindness among working-aged adults globally [1,2,3,4,5]. According to the 2002 American Diabetes Association Position Statement, nearly all patients with type 1 diabetes and over 60% of patients with type 2 diabetes developed retinopathy during the first 20 years of the disease [6]. In 2015, about 1.5 million Americans were diagnosed with diabetes, and an additional 84.1 million Americans had prediabetes [7]. The fast-growing new cases of diabetes suggests that DR will continue to be a major cause of vision loss and associated functional impairment in the U.S. in the coming years.

Because DR can progress to irreversible stages (impossible to restore visual acuity) with relatively few symptoms, early detection and treatment are essential in preventing DR and the subsequent vision loss [8]. Although DR diagnostic and treatment options have significantly advanced over the past decades, the early detection and screening for DR remain challenging due to poor adherence to annual examination guidelines and lack of resources to deploy comprehensive screening programs [3,9], especially in rural/undeveloped areas. Therefore, there is a critical need for stakeholders to research innovative ways that can implement timely, cost-effective detection techniques and/or programs in communities.

Clinical predictive models [10,11,12,13,14] provide an effective, alternative solution to improve the access to early screening for DR under current limitations, by forecasting accurate risk estimates of diseases based on important biomarkers. Predictive models have been extensively investigated and adopted in diabetes studies [15,16,17,18,19,20,21]. In particular for DR, many conditions comorbid with diabetes, such as hyperglycemia, hypertension and dyslipidemia have been found to be significantly associated with DR [1,2,22,23,24,25,26,27]. In addition, HbA1c, fasting plasma glucose, hemoglobin, hematocrit and many other laboratory test values were found to be risk factors for DR development [28,29,30,31,32]. Based on the risk factors identified, a few prediction models were developed to predict the incidence and development of DR [33,34,35,36,37,38]. However, most of the prediction models incorporated a multitude of laboratory variables, leading to no consensus about which laboratory tests are required for effective and economical prediction of DR. The objective of this study is to identify a set of laboratory tests most important for DR prediction, and use the essential laboratory results, in conjunction with other key predictors, to develop an accurate and cost-effective predictive model and an easy-to-use risk index. These predictive technologies can assist healthcare providers in identifying patients at high DR risk and counseling them for ophthalmic examination and proper treatments at early stages.

## 2. Materials and Methods

We performed a retrospective, secondary analysis of electronic health records (EHRs) extracted from two different data sources for derivation and validation, respectively. The details of our data and analysis are elaborated in the remainder of this section.

### 2.1. Data Sources and Data Extraction

We used Cerner Health Facts^®^ EHR data warehouse (Cerner Corporation, Kansas City, MO, USA) as our derivation data source to identify essential predictors, and develop a model and a risk index to predict the onset of DR. Health Facts contains clinical data contributed voluntarily from over 200 hospitals using Cerner EHR systems across the U.S. spanning the past two decades. Cerner Corporation collects and integrates the data with its internally established procedures in compliance with Health Insurance Portability and Accountability Act (HIPAA) laws, thus all data are de-identified. The data in Health Facts^®^ are mostly time-stamped and cover a variety of aspects of patients’ hospital records including encounters, diagnoses, procedures, medications, vital signs, laboratory results, etc. Since Health Facts^®^ has been completely de-identified according to HIPAA regulations, the Institutional Review Boards (IRB) at Oklahoma State University (OSU) exempted the study from review.

In order to validate the predictors and the predictive technologies derived from Health Facts^®^, we used the Healthcare Enterprise Repository for Ontological Narration (HERON) [39] from the University of Kansas Medical Center (KUMC) as our validation data source. HERON was established with the EHRs collected from KUMC and its affiliated clinical organizations since 2010. The data contained in HERON include patient demographics and time-stamped encounter, diagnosis, procedure, laboratory, vital sign and medication records. All the HERON data are de-identified, therefore the validation study was exempted from the IRB review of both OSU and KUMC, and was approved by the HERON Data Request Oversight Committee. It should be noted that KUMC and its affiliated clinical organizations have been using Epic EHR systems since 2007, hence the data included in HERON for validation are completely independent from our derivation cohort extracted from Health Facts^®^.

In this study, we identified diabetic and DR patients from the data sources using corresponding International Classification of Diseases, Ninth and Tenth Revisions, Clinical Modification (ICD-9/10-CM) diagnosis codes, as listed in Table 1. In particular, diabetic patients were defined as having at least one of 250.x, E10.x and E11.x diagnosis codes [21]. Among diabetic patients, those who had 362.0x, E10.31x-E10.35x or E11.31x-E11.35x diagnosis code(s) are identified as DR patients (case). Otherwise, the diabetic patients are considered as non-DR patients (control).

### 2.2. Data Preprocessing

To support developing early prediction models for DR, we employed a window-based data aggregation approach proposed by Ng et al. [40] to preprocess the data. As illustrated in Figure 1, the method first identifies an event of interest (EOI), then assigns two successive time windows prior to the event, namely a prediction window and an observation window. The risk prediction is made at the beginning of the prediction window with the aggregated data in the observation window to estimate the risk of EOI before the actual occurrence of the event. We used the first DR diagnosis as the EOI for the case cohort and selected a random encounter after the diabetes diagnosis as the EOI for the control patients. The lengths of the prediction window and observation window were set to be six months and two years, respectively. (Note that we tested the latest available data before prediction window, and one year and two years for the observation window in our preliminary studies. The two-year observation window resulted in the best predictive accuracy, thus it was chosen).

Our study considered a total of 26 variables including three demographics (gender, race, and the age at the beginning of the prediction window), two additional diabetic microvascular complications—nephropathy and neuropathy (corresponding ICD-9/10 codes are listed in Table 1), duration of diabetes (measured in years from the first diabetic diagnosis to the beginning of the prediction window) and results of 20 different routine blood tests for diabetic patients as listed in Table 2. Before the aggregation of laboratory results, the interquartile range method [41] was utilized to identify and remove outliers. Then, we took the mean to aggregate the laboratory results within the observation window. The two diabetic microvascular complications—nephropathy and neuropathy—were modeled as two binary variables. Specifically, if a complication occurred before the prediction window, the associated variable was marked to be 1, otherwise 0. Furthermore, our analysis only included patients with complete records for all variables; in other words, if a patient record came with any missing demographics or laboratory results (the complication and duration of diabetes variables must not be missing), the record was excluded from our subsequent analysis.

### 2.3. Essential Predictor Identification and Predictive Modeling

As shown in Figure 2, our analysis first evaluated the bivariate association of each variable with the onset of DR. To that end, a Chi-squared test was applied to the categorical variables, including age, gender, race, duration of diabetes and diabetic complications, while the two-sample *t*-test was used for laboratory results. Furthermore, we used the odds ratio (OR) and its 95% confidence interval (CI) to compare the association strengths (with DR) among different levels of each categorical variable. Significant variables (p<0.05) from the bivariate analysis were then selected for the subsequent predictive modeling and key predictor selection.

In order to identify a compact set of predictors with the best predictive power among the variables found statistically significant in the bivariate analysis, we randomly partitioned all the derivation data into a training data set (70%) and a testing data set (30%), and applied the machine-learning-based ensemble predictor selection (EPS) method proposed by Song et al. [42] to our training data set. On the other hand, the testing data set was left out for internal validation. The EPS method consists of two steps: (i) ranking aggregated variable importance and (ii) golden-section search for a minimal predictor count. In the first step, the method builds machine-learning models on bootstrap samples from the training data set, and returns variable importance (in terms of the contribution to the prediction accuracy) for each bootstrap sample. Then, the importance values are aggregated across the bootstrap samples, and the variables are sorted based on the aggregated importance. The second step performs a golden-section search on the sorted variables to determine a minimal set of predictors that can maintain a close predictive accuracy to that given by the full model incorporating all significant variables. In our implementation of the EPS method, we employed extreme gradient boosting (XGBoost) as our machine-learning model. XGBoost is a popular, tree-based machine-learning technology [43], and demonstrated an outstanding performance in EPS [42]. Furthermore, we evaluated the predictive accuracy using the area under the receiver operating characteristic curve (AUC), and used weighted average to aggregate variable importance (the AUC on the bootstrap sample was used as the weight).

### 2.4. Risk Index Development

The machine-learning-based predictive model is a “black box” in nature, thus difficult to interpret [44,45]. To address the issue, risk indices are often developed based on essential predictors to provide predictive tools that are more user-friendly and easier to interpret [46,47]. In the research, we used the scoring method described in [48] to create an index to predict the DR risk. The scoring method consists of the following five steps:(i)Create a logistic regression based on the specified *n* predictors, and obtain the predictors’ coefficients $\left\{\beta_{i}|i=0,1,\cdots ,n\right\}$, where $\beta_{0}$ is the intercept and $\beta_{i}$ is the coefficient of *i*th predictor.(ii)Break down each numerical predictor into intervals (i.e., levels) and determine the reference level for each predictor based on clinical expertise.(iii)Calculate the distance from each level to the reference level in terms of regression risk units for each predictor. The distance is defined as $\beta_{i}\left(M_{ij}-M_{iR}\right)$, where $M_{ij}$ and $M_{iR}$ are the level values of level *j* and the reference level of predictor *i*, respectively. The level value is defined as the middle point for numerical, interval levels and non-negative integers for other types of levels.(iv)Define a constant *B* regarding how many regression risk units can be mapped to one point in the scoring system. In this study, we let $B=5\times \beta_{age}$. In other words, one point in the risk scoring system corresponds to the increased regression risk units associated with a 5-year change in age.(v)Compute the score for each level of a predictor by rounding $\beta_{i}\left(M_{ij}-M_{iR}\right)/B$ to the nearest integer. The risk index is the summation of all predictors’ scores.

All the data preprocessing and predictive technologies presented in this article were implemented using R 3.6.0 [49].

## 3. Results

Figure 3 shows a development workflow for our derivation cohort. In the cohort, we included 3749 DR and 94,127 non-DR diabetic patients (the DR rate is 3.8%). By applying a similar workflow to the validation data source, we obtained a validation cohort with 869 DR and 6448 non-DR diabetic patients (the DR rate is 11.9%). The cohort statistics and bivariate results are listed in Table 2, which shows that the associations of many variables with DR in the validation cohort were consistent with that in the derivation cohort. For example, a longer duration of diabetes is significantly associated with a higher risk of DR in both cohorts (*p*-values <0.001) and gender is a statistically insignificant variable in both cohorts (*p*-values =0.707 and 0.955 in the derivation cohort and the validation cohort, respectively). However, there still exist certain differences in statistics between the two cohorts, especially for the laboratory results. For example, triglycerides (*p*-value =0.436) and an anion gap (*p*-value =0.848) were found to be insignificant in the derivation cohort, but they were significant in the validation cohort (*p*-values =0.003 and 0.011, respectively). Furthermore, chloride, MCHC, bilirubin and WBC were significant variables in the derivation cohort (*p*-values ≤0.001), but they were insignificant in the validation cohort (*p*-values were 0.082,0.084,0.651 and 0.074). Furthermore, an interesting observation from the results is that diabetic patients aged 65 or older have a lower risk of developing DR than their younger peers, and this observation holds for both the derivation and validation cohorts. Though seemly counter-intuitive, a similar finding was reported in [50] (interested readers may refer to the discussion therein for possible reasons, which are beyond the scope of this paper).

We then excluded statistically insignificant variables in the derivation cohort (including gender, triglycerides and anion gap) and entered the remaining 23 variables into the EPS process to build a predictive model and select the essential predictors based on the derivation cohort. The variable importance returned by EPS is shown in Figure 4. Among the 23 variables, 10 were selected as the essential predictors, which include age, creatinine, HbA1c, neuropathy, duration of diabetes, WBC, nephropathy, glucose, hematocrit, and sodium. Comparing the predictive accuracies between the full model that includes all 23 variables and the compact model with only the 10 essential predictors (AUCs are shown in Figure 5), we find that the compact model (AUCs were 0.85 and 0.77 for the derivation and validation cohorts, respectively) achieved very close accuracies to that of the full model (AUCs were 0.85 and 0.78, respectively, for the derivation and validation cohorts) for both the derivation and validation cohorts.

The top 10 essential predictors were entered into a multivariate logistic regression for risk index development. Table 3 lists the logistic regression results as well as the scoring system derived from the results. The risk index calculated based on the scoring system can estimate the DR risk with considerably good accuracy. Its AUC on our derivation cohort was 0.81, and the AUC on the validation cohort was 0.73 (shown in Figure 5), which are very close to that given by the full models. Though the developed risk index ranges from 0 to 160, our results suggest to break down the index into eight intervals with cutoffs: 40, 50, 60, 70, 80, 90, and 100, as shown in Figure 6. From the risk index distributions shown in Figure 6, we can observe that the index is capable of reflecting the trend of the DR risk: as the index score increases, the patient risk of developing DR becomes higher in both the derivation and the validation cohorts, supporting the generalizability of the proposed risk index.

The age-specific AUCs of our compact model and risk index are presented in Table 4. It shows that these technologies had better predictive accuracy for younger patients than for senior patients. Furthermore, there exist certain inconsistencies in the performance between the derivation cohort and the validation cohort for patients aged 85 or older. The phenomenon may be a result of the complex health conditions of senior patients and implies the need to improve the technologies’ accuracy for this group of patients in the future.

## 4. Discussion

A large set of risk factors have been reported to be significantly associated with the incidence of DR in literature, leaving little consensus regarding which variables are essential for DR prediction. A major contribution of our work is to derive and validate a small set of the most important predictors from the large variable set. We showed that these predictors were essential because they contributed a majority of the predictive accuracy and adding more variables did not improve the accuracy significantly.

Our analysis and technologies also have various practical implications. Firstly, we derived a predictive model with a minimum number of predictors that have high prediction accuracy for DR. As previously stated, the current method for detecting or diagnosing DR is the annual ophthalmic exam. However, the annual eye examination has a low compliance rate due to the lack of specialists and high overhead to patients in many areas, especially the rural communities [51]. Our study tackles the poor compliance challenge by developing an automatic predictive model. Its prediction result can provide an effective and early warning for DR risk and trigger practitioners to counsel and refer patients for ophthalmic examination and proper treatments. Comparing with several published models that similarly aim to predict the DR risk, our model includes fewer predictors, resulting in easier interpretation and reduction in the cost related to data collection.

Secondly, the risk index we developed provides a practical and user-friendly tool to monitor the essential predictors for early warning signs of DR. Though the machine learning model achieved high predictive accuracies in both derivation and validation cohorts, its black box nature makes it difficult to understand how decisions are made. Moreover, complex machine-learning algorithms require the support of specific software (such as R or EHR systems) for their execution, which are often less user-friendly to health workers and can lead to additional costs to hospitals/clinics. The proposed risk index was designed to address these issues.

More notably, we externally validated the predictive model and the risk index by testing the technologies on a large patient cohort from an independent EHR database. The technologies demonstrated promising discriminative ability in the external data source, which implies their generalizability to the entire diabetic patient population. As far as we know, this study is the first effort to validate diagnostic predictive models for DR on two distinct patient cohorts.

With respect to how to utilize the predictive technologies, we recommend health workers and patients to use the risk index because of its easy-to-use nature. On the other hand, EHR vendors may integrate the machine-learning model into their EHR systems to make this more accurate prediction tool available to providers.

Caveats: It is worth noting that the predictive technologies are not intended to replace the regular eye examination, which is the gold standard for DR diagnosis. However, they can be useful screening tools to identify those that may be at higher risk to ensure timely diagnosis and intervention. Moreover, it is important to note that the statistical results, presented in this article, reveal the associations between the predictors and the development of DR. The associations do not necessarily reflect the causality. It is still unclear why the laboratory values are significantly associated with DR, and how they contribute to the prediction from the pathological perspective. Therefore, the pathological role of the essential predictors in the development of DR should be investigated in the future.

Future Work: There are several potential directions worth future investigations following this study. Firstly, further investigations on better-quality data of senior diabetic patients are desired to improve the prediction accuracy for this group of patients. Secondly, we plan to include more comorbidities, treatments and laboratory results into the feature selection process and predictive models to find novel, essential predictors as well as improve the predictive accuracy. We are also interested in conducting clinical trials to further validate the essential predictors and the predictive tools. Furthermore, the methods employed in this study can be extended to nephropathy, neuropathy and other diabetic complications for creating a comprehensive prognosis toolkit for diabetes and associated complications. Lastly, the pathological relationship between the development of DR and the key predictors is also an interesting direction for future research.

Limitations: There exist a couple of limitations in the research. Firstly, EHRs do not necessarily capture the complete pictures of patient health. Useful data of a patient may be missing from our EHR data sources for many reasons, such as a patient is not compliant with follow up or treatment, the DR diagnosis is not entered into the EHR problem list due to neglect, and a patient visits other hospitals with different EHR systems. Secondly, due to de-identification, there were a small number (2173) of patients older than 90 were recorded as 90 years old in the derivation cohort (there were no such patients in the validation cohort). The inaccuracy may undermine the prediction performance of our technologies for patients older than 90. More accurate data of patients aged 90 years or older can help address the limitation.

## 5. Conclusions

DR is a major cause of blindness among middle-aged adults over the world. The vision loss that occurs at the late stage of DR cannot be reversed. As a result, diagnosing DR at an early date is crucial. In this study, we conducted a retrospective analysis of EHR data to identify a set of essential predictors of DR. Based on the key predictors, we furthermore derived and validated a compact predictive model and a risk index for early detection of DR. The technologies demonstrated promising accuracies in prediction, both internally on the derivation cohort and externally on the validation cohort. The DR risk given by the predictive technologies can be used as an early warning sign to urge patients to comply with the prompt ophthalmic examination, which has a relatively low compliance rate currently.

## Figures and Tables

**Figure 1 jcm-10-01473-f001:**
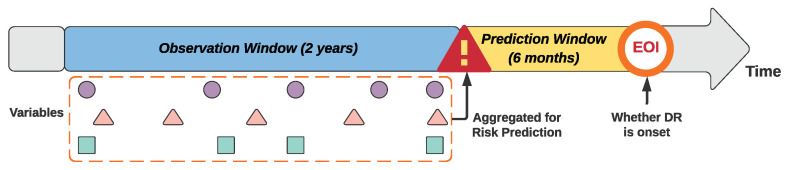
Prediction and observation windows for the predictive modeling.

**Figure 2 jcm-10-01473-f002:**
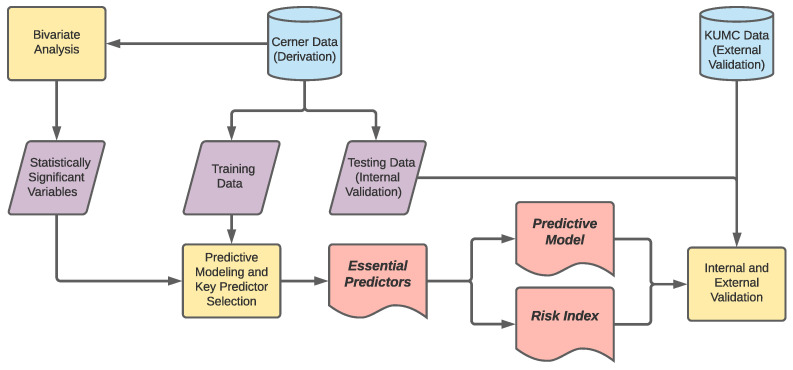
Steps of the derivation and validation analyses.

**Figure 3 jcm-10-01473-f003:**
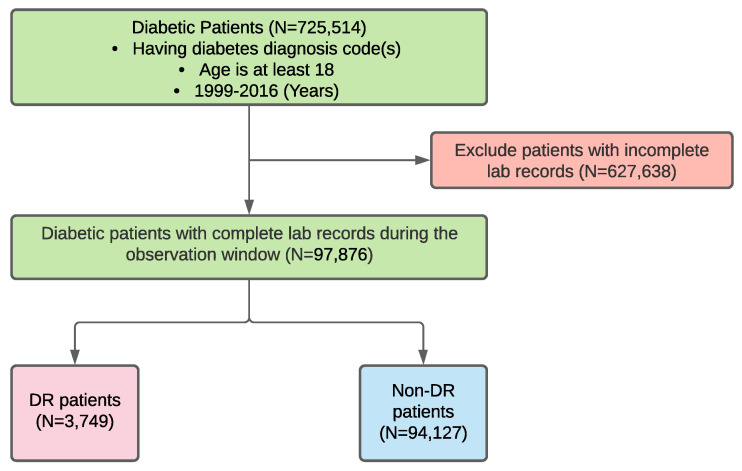
The workflow for the derivation cohort development.

**Figure 4 jcm-10-01473-f004:**
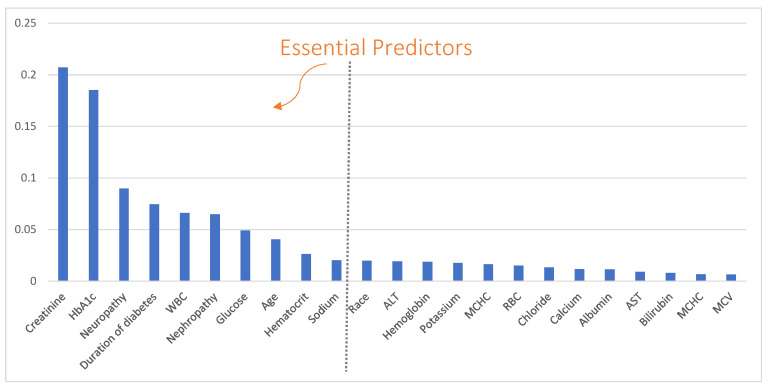
Variable importance and essential predictors returned by the ensemble predictor selection (EPS) method.

**Figure 5 jcm-10-01473-f005:**
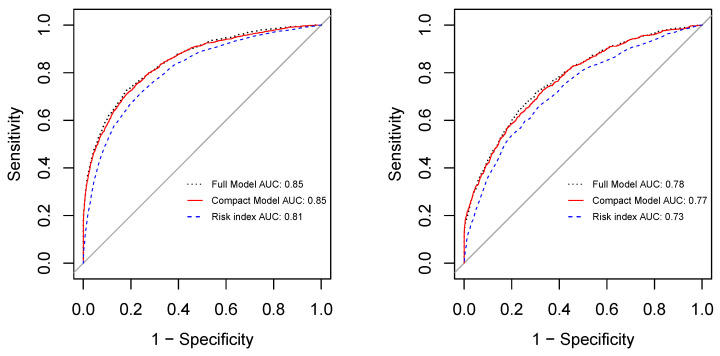
AUCs on the derivation cohort (**left**) and the validation cohort (**right**).

**Figure 6 jcm-10-01473-f006:**
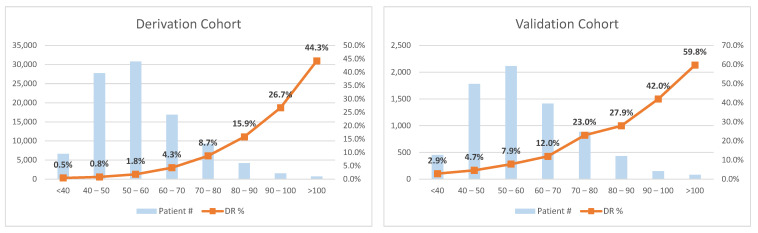
Risk index distributions in the cohorts.

**Table 1 jcm-10-01473-t001:** International Classification of Diseases, Ninth and Tenth Revisions, Clinical Modification (ICD-9/10-CM) codes of diabetes and associated complications.

ICD Version	ICD Code	Code Description
ICD-9-CM	250.x	Diabetes mellitus
	362.0x	Diabetic retinopathy
	250.4x	nephropathy
	250.6x	neuropathy
ICD-10-CM	E10.x	Type 1 diabetes mellitus
	E11.x	Type 2 diabetes mellitus
	E10.31x–E10.35x	Type 1 diabetes mellitus with diabetic retinopathy
	E11.31x–E11.35x	Type 2 diabetes mellitus with diabetic retinopathy
	E10.21	Type 1 diabetes mellitus with diabetic nephropathy
	E11.21	Type 2 diabetes mellitus with diabetic nephropathy
	E10.40	Type 1 diabetes mellitus with diabetic neuropathy
	E11.40	Type 2 diabetes mellitus with diabetic neuropathy

**Table 2 jcm-10-01473-t002:** Statistics and bivariate analytic results of the derivation and validation cohorts.

Demog. & DM	Cerner Derivation Data			KUMC Validation Data		
	# (DR#)	OR (95% CI)	*p*-Value	# (DR#)	OR (95% CI)	*p*-Value
Age			<0.001			0.011
≥85	6036 (75)	Reference	–	179 (7)	Reference	–
18–34	2723 (135)	4.15 (3.12–5.54)	<0.001	261 (37)	4.06 (1.87–10.14)	0.001
35–49	13,247 (620)	3.90 (3.09–5.01)	<0.001	908 (102)	3.11 (1.52–7.48)	0.005
50–64	33,554 (1585)	3.94 (3.14–5.02)	<0.001	2851 (371)	3.68 (1.85–8.71)	0.001
65–74	25,118 (854)	2.80 (2.22–3.58)	<0.001	2024 (238)	3.27 (1.64–7.78)	0.002
75–84	17,198 (480)	2.28 (1.80–2.94)	<0.001	1094 (114)	2.85 (1.41–6.86)	0.008
Gender						
Female	53,396 (2034)	Reference	–	3703 (439)	Reference	–
Male	44,480 (1715)	1.01 (0.95–1.08)	0.707	3614 (430)	1.00 (0.87–1.16)	0.955
Race			<0.001			<0.001
Black	17,993 (1363)	Reference	–	1572 (254)	Reference	–
White	72,488 (2126)	0.37 (0.34–0.40)	<0.001	5005 (499)	0.57 (0.48–0.68)	<0.001
Other	7395 (260)	0.44 (0.39–0.51)	<0.001	1360 (116)	740 (0.76–1.22)	0.768
Duration (years)			<0.001			<0.001
0–1	36,536 (775)	Reference	–	2285 (179)	Reference	–
1–2	26,359 (843)	1.52 (1.38–1.68)	<0.001	1729 (196)	1.50 (1.22–1.86)	<0.001
2–3	13,560 (723)	2.60 (2.34–2.88)	<0.001	1054 (135)	1.73 (1.36–2.19)	<0.001
3–4	8696 (518)	2.92 (2.61–3.27)	<0.001	719 (79)	1.45 (1.09–1.91)	<0.001
>4	12,725 (890)	3.47 (3.14–3.83)	<0.001	1530 (280)	2.64 (2.16–3.22)	<0.001
Nephropathy						
No	92,153 (2711)	Reference	–	6239 (617)	Reference	–
Yes	5723, (1038)	7.31 (6.76–7.90)	<0.001	1078 (252)	2.78 (2.36–3.27)	<0.001
Neuropathy						
No	88,657 (2507)	Reference	–	5986 (559)	Reference	–
Yes	9219 (1242)	5.35 (4.98–5.75)	<0.001	1331 (310)	2.95 (2.53–3.44)	<0.001
**Lab results**	**Case avg (SD)**	**Control avg (SD)**	***p*-value**	**Case avg (SD)**	**Control avg (SD)**	***p*-value**
HbA1c	8.36 (2.03)	7.14 (1.51)	<0.001	8.20 (2.07)	6.95 (1.45)	<0.001
Creatinine	1.94 (1.81)	1.07 (0.46)	<0.001	1.56 (1.45)	1.10 (0.47)	<0.001
Glucose	174.83 (61.95)	142.89 (46.24)	<0.001	170.74 (53.15)	143.78 (40.21)	<0.001
Hemoglobin	12.08 (1.65)	13.03 (1.70)	<0.001	12.40 (1.91)	12.86 (1.90)	<0.001
Hematocrit	36.24 (4.72)	38.97 (4.75)	<0.001	37.36 (5.60)	38.69 (5.58)	<0.001
Calcium	9.12 (0.49)	9.26 (0.44)	<0.001	9.27 (0.49)	9.37 (0.46)	<0.001
Triglycerides	155.56 (89.85)	154.40 (84.71)	0.436	165.96 (94.59)	155.80 (86.11)	0.003
Potassium	4.33 (0.38)	4.24 (0.35)	<0.001	4.20 (0.37)	4.14 (0.32)	<0.001
Chloride	103.00 (3.45)	103.19 (2.85)	0.001	103.11 (3.07)	103.30 (2.69)	0.082
MCH	29.57 (2.00)	29.95 (1.94)	<0.001	29.50 (2.13)	29.83 (2.07)	0.001
Sodium	138.49 (2.39)	138.83 (2.47)	<0.001	137.03 (2.10)	137.31 (2.19)	<0.001
MCHC	33.32 (0.92)	33.43 (0.96)	<0.001	33.19 (0.80)	33.24 (0.78)	0.084
MCV	88.74 (5.31)	89.53 (5.00)	<0.001	88.84 (5.56)	89.67 (5.34)	<0.001
Albumin	3.64 (0.55)	3.86 (0.46)	<0.001	3.80 (0.48)	3.95 (0.46)	<0.001
Bilirubin	0.58 (0.27)	0.61 (0.28)	<0.001	0.60 (0.28)	0.64 (0.30)	0.651
Anion Gap	9.55 (2.69)	9.47 (2.56)	0.058	8.05 (1.83)	7.88 (1.73)	0.011
AST	24.45 (9.79)	24.88 (10.08)	0.009	22.80 (10.28)	24.11 (10.30)	<0.001
ALT	25.18 (12.24)	27.86 (14.47)	<0.001	22.34 (12.23)	25.23 (14.00)	<0.001
RBC	4.10 (0.58)	4.37 (0.56)	<0.001	4.23 (0.66)	4.34 (0.65)	<0.001
WBC	7.97 (2.24)	8.13 (2.21)	<0.001	8.02 (2.35)	8.17 (2.27)	0.074

Demog. and CM: variables related to demographics and status of diabetes mellitus; Duration: duration of diabetes; MCH: mean corpuscular hemoglobin; MCHC: mean corpuscular hemoglobin concentration; MCV: mean corpuscular volume; AST: aspartate transaminase; ALT: alanine transaminase; RBC: red blood cells; WBC: white blood cells.

**Table 3 jcm-10-01473-t003:** Logistic regression results and risk scores for essential predictors.

Variable	Coefficients ($\beta_{i}$)	OR (95% CI)	*p*-Value	Levels	Level Values ($M_{\mathrm{ij}}$)	Points
Age	−0.0187	0.98 (0.97–0.99)	<0.001	18–34	26	12
				35–49	42	9
				50–64	57	6
				65–74	69.5	4
				75–84	79.5	2
				≥85 ${}^{‡}$	87.5	0
Creatinine	0.8601	2.36 (2.24–2.50)	<0.001	<0.5 ${}^{‡}$	0.41	0
				0.5–1	0.75	3
				1–1.5	1.25	8
				1.5–2	1.75	12
				>2	2.68	21
HbA1c	0.2877	1.33 (1.29–1.37)	<0.001	<6 ${}^{‡}$	5	0
				6–8	7	6
				8–10	9	12
				10–12	11	18
				>12	14	28
Neuropathy	0.9229	2.52 (2.27–2.78)	<0.001	No ${}^{‡}$	0	0
				Yes	1	10
Duration of diabetes	0.1455	1.16 (1.13–1.18)	<0.001	<1 ${}^{‡}$	0.5	0
				1–2	1.5	2
				2–3	2.5	3
				3–4	3.5	5
				>4	9.2	14
WBC	−0.1077	0.90 (0.88–0.92)	<0.001	<4	3.5	17
				4–6	5	15
				6–8	7	13
				8–12	10	9
				>12 ${}^{‡}$	18.2	0
Nephropathy	0.5598	1.75 (1.55–1.98)	<0.001	No ${}^{‡}$	0	0
				Yes	1	6
Glucose	0.0059	1.01 (1.00–1.01)	<0.001	<60 ${}^{‡}$	53	0
				60–80	70	1
				80–100	90	2
				100–200	150	6
				>200	364	20
Hematocrit	−0.0609	0.94 (0.93—0.95)	<0.001	<30	25.7	19
				30–35	32.5	15
				35–40	37.5	11
				40–50	45	7
				>50 ${}^{‡}$	55	0
Sodium	0.0822	1.09 (1.07—1.11)	<0.001	<136 ${}^{‡}$	131.5	0
				136–144	140	7
				>144	146.5	13

^‡^ the reference level; *B* = 0.0187 × 5 = 0.0935.

**Table 4 jcm-10-01473-t004:** Age-specific AUCs of predictions.

Models and AUCs		Age Groups					
		18–34	35–49	50–64	65–74	75–84	≥85
Compact Model	Internal AUC	0.92	0.92	0.86	0.81	0.77	0.72
	External AUC	0.86	0.82	0.79	0.73	0.70	0.82
Risk Index	Internal AUC	0.89	0.88	0.83	0.78	0.74	0.65
	External AUC	0.85	0.83	0.74	0.69	0.64	0.72

## Data Availability

Restrictions apply to the availability of Cerner Health Facts^®^ EHR data. The data were obtained from the Cerner Corporation and are available from the authors with the permission of the Cerner Corporation. The HERON data presented in this study are available on request from KUMC. The data are not publicly available because it is proprietary to KUMC.

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
