# Peer review of "Derivation and Validation of Essential Predictors and Risk Index for Early Detection of Diabetic Retinopathy Using Electronic Health Records"

_jcm, 2021, doi:10.3390/jcm10071473_

Round 1

Reviewer 1 Report

The idea of using AI algorithm for identify patients at risk has an important role in medicine and DR can cause a major morbidity and therefor is an important issue.

However as a clinician the suggested model does not add new information for me. Factors such as duration of diabetes, glycemic control and other complications of diabetes are well known as a risk factor for DR, as can be seen at the ADA guidelines for diabetes.

  1. It is not clear whether the cohort is a cohort of primary care patients or patients at a secondary care system. The prevalence of DR is different at different levels of care, and the calculated risk is changing accordingly. It should be stated clearly.
  2. Most of our patients have incomplete data. Moreover incomplete data may indiacte less compliant patients with follow up and treatment, therefore they are at a greater risk for DR. The risk index should be tested for these patients as well; otherwise it is of limited value.

Questions about the risk index itself:

  1. There is no information about microalbumin levels of the patients. Microalbumin can detect nephropathy much earlier than creatinine. Since there is a strong association between diabetic nephropathy and DR this is extremely important.
  2. Patient blood pressure level is not included in the information provide and in the model. Both hypertension and diabetes can cause retinopathy and the combination of them may double the risk.
  3. There is no information about other comorbidities which may increase the risk for DR such as cardiovascular diseases which can also indicate diabetes complications. Also identifying other diseases that might be risk factors for DR in diabetic patients is extremely valuable.

The same question is indicated for various diabetic treatments that may delay or aggravated DR, and may add novel important information for clinicians.

Author Response

Responses to Reviewer 1:

Our team greatly appreciates the reviewer for reviewing our manuscript and providing such insightful feedback. We have read through all the comments carefully. Many of the reviewer’s concerns have actually resulted from the limitations of the data we analyzed in this study. We provided our responses (in purple color) to these concerns in detail. In addition, all modifications according to the responses were made and highlighted in the revised manuscript. 

Comments and Suggestions for Authors

The idea of using AI algorithm for identify patients at risk has an important role in medicine and DR can cause a major morbidity and therefor is an important issue. 

However as a clinician the suggested model does not add new information for me. Factors such as duration of diabetes, glycemic control and other complications of diabetes are well known as a risk factor for DR, as can be seen at the ADA guidelines for diabetes. 

Response: We agree with the reviewer that all the predictors included in our models have been reported in the literature. In fact, in literature there exists a large set of risk factors that are significantly associated with DR. A major contribution of our work is to derive and validate a small set of the most important and necessary predictors from the large variable set. We showed that these predictors are essential because adding more variables does not necessarily increase the predictive accuracy. More importantly, based on the essential predictors, we created and validated a novel risk index that is simple but accurate for DR prediction. Statements with respect to these contributions were also made in our revised manuscript, in lines 198 – 202 and lines 213 – 219:

“A large set of risk factors have been reported to be significantly associated with the incidence of DR in literature, leaving little consensus regarding which variables are essential for DR prediction.  A major contribution of our work is to derive and validate a small set of the most important predictors from the large variable set. We showed that these predictors were essential because they contributed a majority of the predictive accuracy, and adding more variables did not improve the accuracy significantly.”

“Secondly, the risk index we developed provides a practical and user-friendly tool to monitor the essential predictors for early warning signs of DR. Though the machine learning model achieved high predictive accuracies in both derivation and validation cohorts, its black box nature makes it difficult to understand how decisions are made. Moreover, complex machine-learning algorithms require the support of specific software (such as R or EHR systems) to execute, which are often less user-friendly to health workers and can lead to additional costs to hospitals/clinics.  The proposed risk index was designed to address these issues.”

  1. It is not clear whether the cohort is a cohort of primary care patients or patients at a secondary care system. The prevalence of DR is different at different levels of care, and the calculated risk is changing accordingly. It should be stated clearly.

Response: We thank the reviewer for this insightful question. We did not report the level of care in this article for two reasons: (1) Due to the de-identification of hospital information, there is no clear field directly indicating the level of care in our datasets, and (2) The only available data field that might be relevant to care level in our derivation cohort was physician’s “medical specialty”. However, this data filed has more than 58% of records missing. As a result, we decided not to include the level of care in our analysis but focused on the combination of laboratory results and basic risk factors (such as demographics, duration of diabetes, and diabetic complications).   

  1. Most of our patients have incomplete data. Moreover incomplete data may indicate less compliant patients with follow up and treatment, therefore they are at a greater risk for DR. The risk index should be tested for these patients as well; otherwise it is of limited value.

Response: We thank the reviewer for reiterating this intrinsic limitation that commonly exists in retrospective, secondary analysis of EHR data. The data completeness is also a limitation of our study. We report this limitation in our manuscript, in lines 247 – 251, “Firstly, EHRs do not necessarily capture the complete pictures of patient health. Useful data of a patient may be missing from our EHR data sources for many reasons, such as a patient is not compliant with follow up or treatment, the DR diagnosis is not entered into the EHR problem list due to neglect, and a patient visits other hospitals with different EHR systems.”

Questions about the risk index itself:

  1. There is no information about microalbumin levels of the patients. Microalbumin can detect nephropathy much earlier than creatinine. Since there is a strong association between diabetic nephropathy and DR this is extremely important.

Response: We thank the reviewer for suggesting this laboratory result. The reason that we did not include more laboratory results was because some tests were not very complete in our data, especially for the tests that are not blood tests. We have two laboratory tests related to microalbumin in our derivation cohort: (1) Microalbumin/Creatinine Ratio, Urine, and (2) Microalbumin, Urine Random. However, the completeness of the two tests was low. Among the 97,876 patients included in our derivation cohort, there were only 1,820 (<2%) patients who had results of Microalbumin/Creatinine Ratio, and only 36,045 (<37%) patients who had results of Microalbumin, Urine Random. Including microalbumin would let us lose a large proportion of our cohort and may bring bias to the data, thereby reducing the statistical power of our results. Other laboratory results have the same issue; the more tests we included, the less complete the data would be. As a result, we decided not to include microalbumin, or more tests in this study. Data of better quality will be helpful to address this issue. Our team is actively planning prospective studies based on the results and models from this study. We will include microalbumin and more tests in these future investigations.

  1. Patient blood pressure level is not included in the information provide and in the model. Both hypertension and diabetes can cause retinopathy and the combination of them may double the risk.

Response: We thank the reviewer for recommending this vital sign. In our preliminary studies, we did incorporate diastolic and systolic blood pressures, and BMI into our model. However, the AUC of the model had only a minor increase of 0.01. Furthermore, the variable importance of these three vital signs was approximately 0. These vital signs are univariately significant risk factors for DR, but their multivariate predictive powers might be overwhelmed by laboratory results. Therefore, we decided to exclude them from our model, and concentrate on laboratory results.

  1. There is no information about other comorbidities which may increase the risk for DR such as cardiovascular diseases which can also indicate diabetes complications. Also identifying other diseases that might be risk factors for DR in diabetic patients is extremely valuable.

The same question is indicated for various diabetic treatments that may delay or aggravated DR, and may add novel important information for clinicians.

Response: We thank the reviewer for suggesting including more comorbidities and treatments in our analysis. It is in fact one of our future directions to leverage state-of-the-art feature selection approaches to explore all diagnoses and procedures existing in EHRs and identify the most important ones for prediction. The barrier for us to implement the analysis in this study was a large number of complicated diagnosis/procedure codes available in the datasets. There exist 10,081 unique ICD9-/ICD10-CM codes, 2,233 unique ICD9-/ICD10-PCS codes, 2,815 unique CPT codes and 299 unique HCPCS codes in the patient encounters of our derivation cohort. Furthermore, many codes were incomplete (for example, some diabetic diagnoses were recorded using incomplete, unbillable ICD9 code “249”). Since many codes refer to the same disease state or procedures, proper classifications of these codes are needed. However, in current literature there lacks a tool that can efficiently conduct batch classifications for such a large amount of complicated codes. Therefore, we decided to only include the most prevalent diabetic complications in this study, leaving the investigation of more morbidity and treatment related predictors for our future work. This future work statement has been made in our revised manuscript, at lines 239 – 241:

“Secondly, we plan to include more comorbidities, treatments and laboratory results into the feature selection process and predictive models to find novel, essential predictors as well as improve the predictive accuracy.”

Reviewer 2 Report

In this review, the authors explained a cost-effective method to predict diabetic retinopathy at an early stage of the disease. The authors clearly presented different sets of analyses to identify important predictors of DR. The number of patients for both Diabetic and non-diabetic looks reasonable. Also, the limitations of this model are clearly mentioned.

Question 1: In the data-processing section, the authors chose 2-year observation. Is there any rationale why this time course is chosen?

Question 2: which age range do you expect to respond better to this model? is it something that because of more complex diseases in older age, might not be a good predictor for older people? If yes, please describe it in the discussion section

Question 3: Since people with diabetes are prone to nephropathy or neuropathy, is this model working to detect those diseases? If yes, please discuss this in the discussion

There are some spelling and grammatical errors that need to be improved.
